# Shift in G_1_-Checkpoint from ATM-Alone to a Cooperative ATM Plus ATR Regulation with Increasing Dose of Radiation

**DOI:** 10.3390/cells11010063

**Published:** 2021-12-27

**Authors:** Fanghua Li, Emil Mladenov, Rositsa Dueva, Martin Stuschke, Beate Timmermann, George Iliakis

**Affiliations:** 1Institute of Medical Radiation Biology, University Hospital Essen, University of Duisburg-Essen, 45147 Essen, Germany; Fanghua.li@uk-essen.de (F.L.); Emil.Mladenov@uk-essen.de (E.M.); Rositsa.Dueva@uk-essen.de (R.D.); 2Department of Particle Therapy, University Hospital Essen, West German Proton Therapy Centre Essen (WPE), West German Cancer Center (WTZ), German Cancer Consortium (DKTK), 45147 Essen, Germany; Beate.Timmermann@uk-essen.de; 3Division of Experimental Radiation Biology, Department of Radiation Therapy, University Hospital Essen, University of Duisburg-Essen, 45147 Essen, Germany; martin.stuschke@uk-essen.de; 4Institute of Physiology, University Hospital Essen, University of Duisburg-Essen, 45147 Essen, Germany; 5German Cancer Consortium (DKTK), Partner Site University Hospital Essen, German Cancer Research Center (DKFZ), 45147 Essen, Germany

**Keywords:** cell cycle, ionizing radiation, DNA double-strand breaks, checkpoints, DNA end resection, ATM, ATR, DNA-PKcs

## Abstract

The current view of the involvement of PI3-kinases in checkpoint responses after DNA damage is that ATM is the key regulator of G_1_-, S- or G_2_-phase checkpoints, that ATR is only partly involved in the regulation of S- and G_2_-phase checkpoints and that DNA-PKcs is not involved in checkpoint regulation. However, further analysis of the contributions of these kinases to checkpoint responses in cells exposed to ionizing radiation (IR) recently uncovered striking integrations and interplays among ATM, ATR and DNA-PKcs that adapt not only to the phase of the cell cycle in which cells are irradiated, but also to the load of DNA double-strand breaks (DSBs), presumably to optimize their processing. Specifically, we found that low IR doses in G_2_-phase cells activate a G_2_-checkpoint that is regulated by epistatically coupled ATM and ATR. Thus, inhibition of either kinase suppresses almost fully its activation. At high IR doses, the epistatic ATM/ATR coupling relaxes, yielding to a cooperative regulation. Thus, single-kinase inhibition suppresses partly, and only combined inhibition suppresses fully G_2_-checkpoint activation. Interestingly, DNA-PKcs integrates with ATM/ATR in G_2_-checkpoint control, but functions in its recovery in a dose-independent manner. Strikingly, irradiation during S-phase activates, independently of dose, an exclusively ATR-dependent G_2_ checkpoint. Here, ATM couples with DNA-PKcs to regulate checkpoint recovery. In the present work, we extend these studies and investigate organization and functions of these PI3-kinases in the activation of the G_1_ checkpoint in cells irradiated either in the G_0_ or G_1_ phase. We report that ATM is the sole regulator of the G_1_ checkpoint after exposure to low IR doses. At high IR doses, ATM remains dominant, but contributions from ATR also become detectable and are associated with limited ATM/ATR-dependent end resection at DSBs. Under these conditions, only combined ATM + ATR inhibition fully abrogates checkpoint and resection. Contributions of DNA-PKcs and CHK2 to the regulation of the G_1_ checkpoint are not obvious in these experiments and may be masked by the endpoint employed for checkpoint analysis and perturbations in normal progression through the cell cycle of cells exposed to DNA-PKcs inhibitors. The results broaden our understanding of organization throughout the cell cycle and adaptation with increasing IR dose of the ATM/ATR/DNA-PKcs module to regulate checkpoint responses. They emphasize notable similarities and distinct differences between G_1_-, G_2_- and S-phase checkpoint regulation that may guide DSB processing decisions.

## 1. Introduction

Exposure of cells to ionizing radiation (IR) induces DNA double-strand breaks (DSBs) that cause cell death and cancer [1,2,3,4]. DSBs activate in cells of higher eukaryotes a complex network of signaling pathways known as the DNA damage response (DDR) that senses DSBs and coordinates their repair with the cellular metabolism, aiming to safeguard genomic stability—or alternatively, to eliminate irreparably damaged cells by inducing apoptosis or senescence [5,6,7]. Four repair pathways: classical non-homologous end-joining (c-NHEJ), homologous recombination (HR), alternative end-joining (alt-EJ) and single-strand annealing (SSA) are implicated in DSB repair [2,8,9].

Typically, ataxia–telangiectasia mutated (ATM), a member of the phosphoinositide 3-kinase (PI3K)-related family of protein kinases (PIKKs) is placed at the apex of DDR, owing to its involvement in the initial phosphorylation of H2AX [10,11,12,13]. Two additional members of the PIKK family, DNA-dependent protein kinase catalytic subunit (DNA-PKcs) and ATM and RAD3 related (ATR), serve distinct functions in c-NHEJ and HR, respectively [12]. The activation of all three kinases is intimately linked to their recruitment to DSBs through dedicated sensor proteins and protein complexes, MRN, KU70/80 and ATRIP, respectively, that causes their activation and mediates the phosphorylation of numerous, frequently overlapping, substrates [12,14,15]. While there is ample evidence for the function of ATM, ATR and DNA-PKcs independently of each other in distinct aspects of DDR and DSB repair [12], crosstalk and collaboration are also known and presumably serve to optimize DSB processing in ways that remain to be elucidated in their details.

A central function of DDR is the activation of checkpoints that delay cell cycle progression and provide time for repair before transition to the next phase, which may compromise repair by altering the context of the DSB in chromatin. In higher eukaryotes, IR-induced DSBs activate checkpoints in the G_1_-, S- and G_2_ phase of the cell cycle [8,12,16]. While ATM is implicated in checkpoints activated in the G_1_-, S- or G_2_ phase, ATR is implicated only partly in the S- and G_2_ phase checkpoints, and DNA-PKcs is not considered a regulatory component of checkpoint response.

Recently, we reported a detailed analysis of the G_2_ checkpoint in cells exposed to IR in the G_2_ phase of the cell cycle. By strictly focusing on the cohort of G_2_-phase irradiated cells, we uncovered intriguing aspects of a crosstalk between DNA-PKcs, ATM and ATR and demonstrated a striking dependence of this crosstalk on the IR dose [17,18]. Specifically, the G_2_-checkpoint activation in cells exposed to low IR doses (defined here as doses below 2 Gy), is epistatically regulated by ATM and ATR, with inhibition of either kinase almost fully suppressing activation. Notably, when G_2_-phase cells are exposed to high (above 2 Gy) IR doses, the tight ATM/ATR coupling relaxes and independent outputs to the G_2_ checkpoint occur so that individual inhibition of ATM or ATR only partly suppresses checkpoint activation. Indeed, the combined inhibition of both kinases is required to fully suppress the activation of this checkpoint [18]. Interestingly, in cells irradiated in the G_2_ phase, DNA-PKcs integrates with ATM and ATR in checkpoint regulation but functions as a strong suppressor of checkpoint hyperactivation by enabling checkpoint recovery, both at high and low IR doses [18]. Notably, DNA end resection at DSBs is regulated in a completely parallel manner and feeds into the activation of ATR that here is also able to regulate resection [18].

It is particularly notable and relevant for the present work that the crosstalk between DNA-PKcs, ATM and ATR in the regulation of resection and the G_2_ checkpoint changes profoundly when cells are irradiated in the S phase and analyzed after they progress to the G_2_ phase [17]. In this case, the G_2_ checkpoint is activated exclusively by ATR through DSB end resection, similarly at high and low IR doses. Resection is negatively regulated by DNA-PKcs and ATM in an epistatic manner, and ATR seems now to respond to, but not to regulate, resection [17]. Consistently, in S-phase irradiated cells, DNA-PKcs and ATM couple to regulate G_2_-checkpoint recovery [17].

These results integrate DNA-PKcs, ATM and ATR in a module, which facilitates their crosstalk through their spatial and temporal organization at the DSB. It is our working hypothesis that this crosstalk serves to integrate inputs from the type of DSBs, the chromatin environment and the phase of the cell cycle to determine initial repair pathway engagement, and in the case of failure, the coordination of backup processing options [17,18]. The observation that the type of DNA-PKcs/ATM/ATR crosstalk changes profoundly depending upon whether cells are irradiated in the G_2_- or S phase, at high or low IR doses, or with high or low LET radiation [19], adds new dimensions to our understanding of checkpoint response and suggests that studies using similar approaches for cells irradiated in other phases of the cell cycle are warranted.

Therefore, here we investigate the organization and modular integration of DNA-PKcs, ATM and ATR, in the regulation of the G_1_ checkpoint in cells irradiated either in the G_0_- or G_1_ phase of the cell cycle. G_0_/G_1_ are particularly interesting cell cycle phases to investigate such interdependencies, as cells lack a sister chromatid and are therefore unable to carry out HR, and have low CDK activity and C-terminal binding protein 1 interacting protein (CtIP) levels, which suppress resection and, thus, ATR activation [8,12]. Furthermore, the regulation of cell transition from the G_1_- or G_0_- to the S phase was extensively investigated in the past and is currently enriched by novel discoveries based on single-cell observations [20,21,22,23,24,25,26].

We report that in cells exposed to low IR doses in the G_0_/G_1_ phase, ATM is practically the sole regulator of the G_1_ checkpoint. At high IR doses, ATM remains dominant, but ATR now measurably contributes to the checkpoint. At high IR doses, limited DSB end resection is observed that is suppressed by the combined inhibition of ATM and ATR. DNA-PKcs fails to exert consistent effects on the regulation of the G_1_ checkpoint in this type of experiment.

## 2. Materials and Methods

### 2.1. Cell Culture and Irradiation

Cells were grown at 37 °C in a humidified atmosphere of 5% CO_2_ in air and were selected to represent normal tissue and tumor origins. The immortalized, normal human fibroblast 82-6 hTert cells were maintained in Earl’s Minimum Essential Medium (MEM), supplemented with 10% fetal bovine serum (FBS) and 1% non-essential amino acids. The lung cancer cells, A549, were maintained in McCoy’s 5A medium, supplemented with 10% FBS. Serum deprivation (SD) was employed to generate cultures of cells highly enriched in the G_1_/G_0_ phases of the cell cycle. To this end, for both 82-6 hTert and A549 cells, 0.1 million cells were plated per 60 mm dish and allowed to grow for 2 days. Subsequently, cells were transferred to serum-free medium (procedure designated: serum deprivation, SD) and incubated for two more days before use in experiments.

Cells were exposed to IR at room temperature (RT) using a 320 kV X-ray machine with a 1.65 mm Al filter (GE Healthcare). The dose rates at 500 and 750 mm distance from the source were 3.2 Gy/min and 1.4 Gy/min, respectively. A rotating irradiation table ensured even dose distribution in a defined ring of the irradiation field, where dishes were placed and which showed less than 5% radial fluctuation in radiation intensity. Different doses were used in different experiments in the range between 1 and 10 Gy. Details are provided in individual experiments.

### 2.2. Pyronin and Ki67 Staining

Pyronin/Hoechst staining was used to characterize G_0_ cells. For each cell line, 1 × 10^6^ cells were harvested and washed with 10 mL phosphate buffered saline (PBS) by spinning down for 5 min at 200× *g*, room temperature (RT). Cells were resuspended in 0.5 mL PBS and fixed for at least 2 h by adding 4.5 mL pre-chilled (−20 °C) 70% ethanol, dropwise, while vortexing to avoid clumping. For staining, ethanol was removed by centrifugation and cells were washed twice with 5 mL FACS buffer (1% BSA, 1 mM EDTA in PBS) supplemented with 0.5 µL (2U) RNase inhibitor/mL (New England Biolabs, Ipswich, MA, USA, M0314) and stained in 0.5 mL Hoechst/PY staining solution (FACS buffer containing 2 µg/mL Hoechst 33,342 (Thermo Fisher Scientific, Waltham, MA, USA, H1399), and 4 µg/mL pyronin (Sigma-Aldrich, St. Louis, MO, USA, P9172)) and analyzed 1 h later in a flow cytometer (Gallios, Beckman-Coulter, Brea, CA, USA). Alternatively, Ki67 was used for the same purpose. Then, 1 × 10^6^ cells were harvested and washed once with 1 mL PBS. Cells were fixed by incubating in PFA solution (3% paraformaldehyde, supplemented with 2% sucrose in PBS) for exactly 10 min at RT; after washing once using 1 mL PBS, the fixed cells were permeabilized by incubating in 0.5% Triton X-100 in PBS for 10 min at RT. Subsequently, cells were blocked by incubating in PBG (0.2% skin fish gelatin, 0.5% BSA fraction V, in PBS) overnight at 4 °C. For staining, cells were incubated for 1.5 h with a primary antibody against Ki67 (Abcam, Cambridge, UK, ab15580), and 1 h with secondary antibody conjugated with AlexaFluor 488 (Thermo Fisher Scientific, A11034). Finally, cells were stained with 40 μg/mL propidium iodide (PI, Sigma-Aldrich). Analysis was carried out by flow cytometry and was followed by quantification (Kaluza 1.3; Gallios, Beckman Coulter, Pasadena, CA, USA).

### 2.3. G_1_-Checkpoint Determination

To measure the IR-induced checkpoint activation in the G_1_ phase, exponentially growing cultures, or cultures of cells enriched in G_0_ by SD were incubated in complete growth medium and the reduction in the fraction of cells with G_1_ DNA content followed as a function of time after IR. To enable the use of this parameter as a gauge for the G_1_ checkpoint, cell cultures were supplemented, just before IR, with 40 ng/mL nocodazole. Nocodazole prevents the division of cells that reach metaphase and thus, cells from entering the G_1_ phase compartment. As a result, progression through the cell cycle of cells in the G_1_ phase at the time of nocodazole addition can be measured by simply following the reduction in the percentage of G_1_ cells. The duration of the G_1_ checkpoint is then conveniently measured as the IR-induced delay in the reduction of the G_1_ fraction. For analysis, cells were stained with 40 µg/mL PI at RT for 15 min and measured in a flow cytometer (Gallios, Beckman Coulter).

### 2.4. Treatment of Cells with Kinase Inhibitors

2-Morpholin-4-yl-6-thianthren-1-yl-pyran-4-one (KU55933, ATM inhibitor, to be termed here ATMi, Calbiochem; IC_50_ ATM  =  13 nM; IC_50_ ATR  =  100 μM; IC_50_ DNA-PKcs  =  2.5 μM) was dissolved in DMSO (Sigma-Aldrich) at 10 mM and used at 10 μM final concentration. 8-(4-Dibenzothienyl)-2-(4-morpholinyl)-4H-1-benzopyran-4-one (NU7441, DNA-PKcs inhibitor, to be termed here DNA-PKi, Tocris Bioscience; IC_50_ DNA-PKcs  =  13 nM; IC_50_ ATM  =  100 μM; IC_50_ ATR  =  100 μM) was dissolved in DMSO at 10 mM and used at 2.5 μM final concentration. 3-Amino-6-[4-(methylsulfonyl)phenyl]-N-phenyl-2-pyrazinecarboxamide (VE821, ATR inhibitor, to be termed here ATRi, Haoyuan Chemexpress; IC_50_ ATR  =  26  nM; IC_50_ ATM >8  μM; IC_50_ DNA-PKcs  =  4.4  μM) was dissolved in DMSO at 10  mM and used at 5  μM final concentration. (3R, 4S)-rel-4-[[2-(5-fluoro-2-hydroxyphenyl)-6,7-dimethoxy-4-quinazolinyl]amino]-alpha,alpha-dimethyl-3-pyrrolidinemethanol (CCT241533, termed here CHK2 inhibitor I, CHK2i.1, Medchemexpress; IC_50_ CHK2 = 3nM, IC_50_ CHK1 = 245 nM) was dissolved in DMSO at 1 mM and used at final concentration of 0.3 or 1  μM. 2-(4-(4-Chlorophenoxy)phenyl)-1H-benzimidazole-5-carboxamide (BML277, termed here CHK2 inhibitor II, CHK2i.2, Merckmillipore; IC_50_ CHK2 = 15 nM, IC_50_ Cdk1 = 12 μM, IC_50_ CK1 = 17 μM) was dissolved in DMSO at 1 mM and used at a final concentration of 1 µM. 2-Bromo-N-[(2-bromoethylamino)-[(3-methyl-2-nitroimidazol-4-yl)methoxy]phosphoryl]ethanamine (PF47736, CHK1 inhibitor, termed here CHK1i, Selleckchem; IC_50_ CHK1 = 0.49 nM; IC_50_ VEGFR2 = 8 nM; and IC_50_ CHK2 = 47 nM) was dissolved in DMSO at 1 mM and used at a final concentration of 0.5 µM. 5-(3-Fluorophenyl)-3-ureidothiophene-2-carboxylic acid N-[(S)-piperidin-3-yl]amide (AZD7762, dual CHK1 + CHK2 inhibitor, to be termed here CHK1-2i, Tocris Bioscience; IC_50_ CHK1 = 5 nM; IC_50_ CHK2 = 5 nM) was dissolved in DMSO at 1 mM and used at a final concentration of 0.5 µM. All inhibitors were administered 1 h before irradiation and were maintained until collection of cells for analysis.

### 2.5. Analysis of DNA End Resection at DSBs by Flow Cytometry

For DNA end resection analysis using RPA70 detection, we employed previously described methods [17,18,27]. Exponentially growing cells were pulse labeled for 30 min with 5 µM 5-ethynyl-2′-deoxyuridin (EdU). After treatment with EdU, the growth medium was removed, and cells were rinsed once with pre-warmed PBS, returned to the growth medium and exposed to X-rays. At different times thereafter, cells were collected by trypsinization and unbound RPA extracted by incubating cells for 2 min in ice-cold PBS containing 0.2% Triton X-100. Cells were spun down for 5 min and fixed for 15 min in PFA solution. Cells were blocked with PBG buffer overnight at 4 °C and incubated for 1.5 h with a monoclonal antibody raised against RPA70 [28]. Cells were washed twice with PBS and incubated for 1.5 h with a secondary antibody conjugated with AlexaFluor 488. Subsequently, the EdU signal was developed using an EdU staining kit according to the manufacturer’s instructions. Finally, cells were stained with 40 μg/mL PI. Cell cycle specific analysis was carried out using flow cytometry combined with quantification by Kaluza 1.3, as described earlier [27].

### 2.6. Indirect Immunofluorescence (IF) and Image Analysis

For immunofluorescence analysis, cells were grown on poly-L-lysine (Biochrom, Cambridge, UK) coated coverslips. Generally, the procedure is similar to the analysis of DNA end resection by flow cytometry. Specifically, unbound RPA was extracted using 0.5% Triton X-100 in ice cold PBS for 15 min. DNA content was determined through staining with 200 ng/mL diamidin-2-phenylindol (DAPI, Thermo Fisher Scientific), instead of PI, and coverslips mounted with PromoFluor antifade reagent (PromoCell, Heidelberg, Germany). Scanning was carried out on AxioScan Z1 (Zeiss, Oberkochen, Germany). In each slide, 4 fields were scanned (corresponding to about 15,000 nuclei) and processed using image analysis software (Imaris, Bitplane, Zürich, Switzerland).

### 2.7. Western Blotting

Cells were collected and washed twice in ice-cold PBS. Approximately 5 × 10^6^ cells were lysed for 30 min in 0.2–0.5 mL of ice-cold RIPA buffer (Thermo-Fisher) supplemented with Halt^TM^ phosphatase and protease inhibitor cocktails, as recommended by the manufacturer. Lysates were spun down for 15 min at 12,000× *g*, 4 °C and the protein concentration was determined in the supernatants using the Bradford assay. Standard protocols for SDS-PAGE and immunoblotting were employed. Unless otherwise indicated, 50 μg RIPA whole cell extract was loaded in each lane. The transfer of proteins onto nitrocellulose membranes and incubation with primary/secondary antibodies were performed according to standard procedures. The primary antibodies used were: anti-p53-pS15 (Cell Signaling Technology, Danvers, MA, USA, 9284), anti-Cyclin D1 (Cell Signaling Technology, 2922), anti-CHK2 (Santa Cruz Biotechnology, Dallas, TX, USA, sc-9064), anti-CHK2-pT68 (Cell Signaling Technology, 2661), anti-CHK2-pS516 (Cell Signaling Technology, 2669), anti-CHK1 (Santa Cruz Biotechnology, sc-8408), anti-CHK1-pS345 (Cell Signaling Technology, 2341), anti-CHK1-pS296 (Cell Signaling Technology, 2349), anti-GAPDH (UBP Bio, Dallas, TX, USA, Y1041). The secondary antibodies were anti-mouse IgG or anti-rabbit-IgG conjugated with IRDye680 or IRDye800 (LI-COR Biosciences, Lincoln, NE, USA, 92668020, 92632210 and 92632210). Immunoblots were visualized by scanning membranes in an infrared scanner (Odyssey, Li-COR Biosciences). Western blots were processed by using the brightness and contrast functions of the Odyssey software.

### 2.8. Statistical Analyses

Results are expressed as mean ± standard error (SE). Statistical significance between experimental groups were determined by *t*-test. The significance of differences between individual measurements is indicated by a symbol: * *p* < 0.05, ** *p* < 0.01, N. S. non-significant.

## 3. Results

### 3.1. ATM Is the Main Regulator of G_1_ Checkpoint in Cells Irradiated in G_1_ Phase

To study the interplay between ATM, ATR and DNA-PKcs in the regulation of G_1_-checkpoint, we employed an immortalized, non-transformed human fibroblast cell line, 82-6 hTert that expresses wild-type p53 and manifests a robust G_1_ checkpoint. To assess G_1_-checkpoint activation, we irradiated actively growing cells and followed by flow cytometry changes in the fraction of cells in G_1_ phase in the presence of nocodazole. Under these growth conditions, the G_1_ checkpoint can be assessed from the decrease in the fraction of cells in the G_1_ phase. Indeed, in non-irradiated cell populations, progression through the cell cycle depletes the G_1_ compartment (Figure 1a,b) from ~70% to ~20% in 24 h, as cells are arrested by nocodazole in the G_2_/M phase (Figure 1a,b). Nocodazole has no detectable effects on the progression of G_1_ cells into the S phase. Notably, exposure to a low IR dose of 1 Gy markedly delays the normal progression of G_1_ cells into the cell cycle, with more than 60% of cells remaining in G_1_ at 24 h (Figure 1a,b and Appendix A). While progression of non-irradiated G_1_-phase cells into the cell cycle varied slightly between experiments, the IR-induced delay was always pronounced and clearly measurable (compare 0 Gy data in Appendix A). We conclude that this delay is a manifestation of the activation of a checkpoint in the G_1_ phase, specifically in cells irradiated in the G_1_ phase.

To investigate the role of ATM and ATR in the manifestation of this checkpoint, we treated cells with small molecule inhibitors of the kinases themselves, as well as of their downstream effector kinases, CHK2 and CHK1, respectively. Treatment with ATMi (Figure 1b) causes irradiated cells to exit the G_1_ phase with kinetics identical to those of ATMi-treated unirradiated cells (Appendix A), suggesting complete abrogation of the checkpoint. This checkpoint is not implemented by CHK2, as two different inhibitors of this kinase, CHK2i.1 and CHK2i.2, administered at 1 µM, fail to detectably suppress its activation (Appendix A). Notably, Appendix A shows that CHK2i.2 at 1 µM fully suppresses the autophosphorylation of CHK2 at pS516 that is required for activity. At the same time, phosphorylation of CHK2 by ATM at pT68 remains unchanged and documents that CHK2 is activated by ATM. As expected, ATMi effectively suppresses CHK2 phosphorylation by ATM at pT68 that is essential for its activation, and as a consequence, also the autophosphorylation at pS516 (Appendix A). Appendix A shows that also CHK2i.1, administered at 0.3 or 1 µM, completely suppresses CHK2 kinase activity, as indicated by the absence of pS516 autophosphorylation. Thus, CHK2 fails to regulate the G_1_ checkpoint, despite its canonical activation.

Treatment with ATRi has also no effect on the G_1_ checkpoint (Figure 1b and Appendix A) and combined treatment with ATMi + ATRi generates an effect equivalent to ATMi alone (Appendix A). The lack of ATR contribution to the G_1_ checkpoint at low IR doses is further underscored by the ineffectiveness of CHK1i, a highly specific inhibitor of CHK1 [29], as well as by CHK1-2i (Appendix A), an inhibitor of both CHK1 and CHK2 [30].

Surprisingly, treatment of unirradiated cells with DNA-PKcsi, in the presence or absence of ATMi and/or ATRi, inhibits their normal progression through the cell cycle (Appendix A), which precludes analysis of the G_1_ checkpoint at this low IR dose (Appendix A). An adverse effect of DNA-PKcsi was also reported for rodent G_1_ cells [31] and may reflect an as of yet uncharacterized contribution of DNA-PKcs to normal cell cycle progression.

We reported that the crosstalk between ATM and ATR in the regulation of the G_2_ checkpoint in cells irradiated in G_2_ changes with IR dose: from purely epistatic regulation at low IR doses to partly independent but cooperative at high IR doses [17,18]. Therefore, we investigated whether similar effects manifest with increasing IR dose in the regulation of the G_1_ checkpoint. After exposure of 82-6 hTert cells to 10 Gy, we observed a strong checkpoint that almost entirely prevents the release of cells from the G_1_ phase in the 48 h of observation (Figure 1c). Notably, checkpoint activation at high IR doses still requires the activity of ATM, as treatment with ATMi causes the release from the G_1_ phase of a considerable fraction of cells. However, at this high IR dose, the suppression of the G_1_ checkpoint by ATMi is incomplete, and a substantial fraction of cells remains arrested in the G_1_ phase, indicating residual checkpoint activity (Figure 1c).

ATRi alone has at this high IR dose a small effect on the G_1_ checkpoint (Figure 1d). Strikingly, however, simultaneous treatment with ATMi and ATRi causes its complete suppression, particularly when considering inhibitor effects on non-irradiated cells (Figure 1e). Notably, at 10 Gy, inhibition of CHK1 has an effect on the G_1_ checkpoint that is equivalent to that of ATRi (Figure 1d), and combined ATMi/CHK1i treatment completely suppresses the checkpoint with less toxicity than combined ATMi/ATRi treatment (Figure 1e). Surprisingly, even at high IR doses, the inhibition of CHK2 with either inhibitor has no effect on the G_1_ checkpoint (Appendix A). DNA-PKcsi exerts no effect on the G_1_ checkpoint under these experimental conditions but suppresses again the normal progression of unirradiated G_1-_ cells into the cell cycle (Figure 1f).

We conclude that in cycling G_1_ cells exposed to low IR doses in the G_1_ phase, activation of the G_1_ checkpoint only depends on ATM. However, at high doses, ATR and its downstream effector CHK1 clearly contribute, and dual inhibition is required to achieve full checkpoint suppression. Contributions from DNA-PKcs under the experimental conditions employed here are not discernible, and adverse effects of DNA-PKcsi on the progression of non-irradiated G_1_ cells into the S phase are noted.

### 3.2. Regulation of G_1_ Checkpoint in G_0_-Irradiated Cells

We designed experiments to study G_1_-checkpoint activation specifically in cells irradiated in the G_0_ phase of the cell cycle, as we extensively study cell radiosensitivity and DSB processing (particularly alt-EJ) throughout the cell cycle and in the G_0_ phase [3,31,32,33,34,35,36,37]. Figure 2a shows the growth characteristics of 82-6 hTert cells under normal growth conditions and following SD after two days, using the protocol outlined under “Material and Methods”. The cell cycle distributions at the indicated time points show that under both growth conditions, cells accumulate with a G_1_-phase content, as they stop dividing and enter a plateau phase (Figure 2b). The staining of cells for Ki67 (Figure 2c), a widely used proliferation marker [38,39,40], shows that under both growth conditions, cells become predominantly Ki67 negative as they reach plateau, which we interpret as evidence that they have entered the G_0_ phase. Additionally, analysis of RNA content using pyronin Y/Hoechst 33342 double staining (Figure 2d) [41,42] shows lower levels, particularly for SD cells after 6 days of growth. Hence, we refer to plateau-phase cells obtained using either protocol as G_0_ cells. Although both growth conditions generate cultures highly enriched in G_0_ cells, in the experiments described below, we exclusively use SD cultures. This is mainly because SD cultures reach plateau phase at low cell numbers, which permits induction of proliferation by transfer to complete growth medium. Thus, assessment of the G_1_ checkpoint is possible without trypsinization and re-platting of cells at low numbers that, by itself, would generate extra stress, confounding the analysis.

To analyze G_1_-checkpoint activation in cells irradiated in G_0_, SD cultures of 82-6 hTert cells were irradiated and transferred to complete growth medium supplemented with nocodazole. In non-irradiated controls, transfer to growth medium stimulates progression through the cell cycle, indicated by the increase in the fraction of S-phase cells 16 h later (Figure 3a). Owing to the presence of nocodazole, the G_1_ compartment becomes depleted with progressing time and the G_2_ compartment becomes filled (Figure 3a). Indeed, the fraction of cells in the G_1_ phase drops to ~50% at 24 h and reaches ~10% at 48 h (Figure 3b). This response demonstrates progression through the cell cycle of the majority of cells in SD cultures after transfer to fresh growth medium and validates the approach for the G_1_-checkpoint analysis in cells irradiated in G_0_. Analysis of Ki67 and pyronin Y after growth stimulation shows that the proportion of negative cells for these proliferation markers decreases after medium change with kinetics similar to those of cell entry into the S phase (Figure 3a–d).

Notably, exposure of cells to 1 Gy of IR strongly delays cell release from the G_1_ phase in the 48 h of observation, with more than 60% of cells arrested in this phase (Figure 3a,b). This is direct evidence for the activation of a robust checkpoint in G_1_, when cells are irradiated in G_0_. Exposure to 4 Gy prevents almost completely cell release from the G_1_ phase and further dose escalation to 10 or 20 Gy has only a small additional effect. Notably, parallel analysis of Ki67 signal shows that cells remain negative for this marker during the checkpoint (Figure 3c). The pyronin Y signal, on the other hand, shows no clear modulation after IR, suggesting that the regulation of processes responsible for the observed reduction occur even during the checkpoint (Figure 3d).

While ATMi has no detectable effect on the progression of unirradiated 82-6 hTert G_0_ cells into the cell cycle, it completely suppresses the delay in the release of cells from the G_1_ phase, imposed by the checkpoint after exposure to 1 Gy (Figure 4a). CHK2i.1 and CHK2i.2 are again without detectable effect (Figure 4b) despite the fact that CHK2 is activated in an ATM-dependent manner, as expected, and both inhibitors suppress CHK2 autophosphorylation at pS516 at the concentrations used (Appendix A). As for cells irradiated in the G_1_ phase, ATRi has no effect on the G_1_ checkpoint in cells irradiated in G_0_ (Figure 4c). Consistently, inhibition of CHK1 with two different inhibitors shows no detectable effect (Figure 4c). Treatment with DNA-PKcsi, alone or in combination with ATMi or ATRi, variably suppresses the release of unirradiated cells from G_1_, as shown in Figure 4d. In cells exposed to 1 Gy, DNA-PKcsi, alone or in combination with ATRi, completely suppresses the release of cells from G_1_, while it fails to modulate the checkpoint suppression by ATMi, when administered alone or in combination with ATRi (Figure 4e). We conclude that also in cells exposed in the G_0_ phase to low doses of IR, the G_1_ checkpoint is regulated predominantly by ATM, but not through CHK2.

Exposure of G_0_ 82-6 hTert cells to 4 or 10 Gy generates the strong G_1_ checkpoint response shown in Figure 3b and Figure 5a,b. At these high IR doses, inhibition of ATM only partly suppresses checkpoint activation. Strikingly, at these IR doses, the inhibition of ATR and CHK1 also generates a small effect on the checkpoint (Figure 5c,d), and combined ATMi + ATRi or ATMi + CHK1i treatment allows irradiated G_0_ cells to progress into the cell cycle with kinetics similar to those of their unirradiated counterparts exposed to inhibitors, while CHK2 inhibition is ineffective again (Figure 5e,f). This result indicates the complete abrogation of the G_1_ checkpoint and is similar to that observed in cells irradiated in the G_1_ phase. At 4 Gy, DNA-PKcsi leaves the G_1_ checkpoint unchanged (Appendix A), while it suppresses the checkpoint inhibition mediated by ATM (Appendix A) or ATR (Appendix A). Cells treated with a combination of all three inhibitors progress through the cycle with kinetics similar to those of their unirradiated counterparts (Appendix A).

Experiments with A549 cells, a human tumor cell line, show similar patterns of growth and cell cycle distribution as 82-6 hTert cells, either when grown in a complete medium or following SD (Appendix A). However, A549 G_0_ cells obtained by SD progress after medium change faster through the cell cycle than 82-6 hTert cells, and display a larger proportion of G_2_ phase cells, 32 h later (Appendix A). Exposure to 1 Gy activates in these cells as well a checkpoint in the G_1_ phase, but of lower effectiveness (Appendix A). Yet, dose escalation to 4 Gy induces a very strong checkpoint, and further escalation to 10 or 20 Gy completely prevents the release of G_1_ cells into the cell cycle (Appendix A). G_0_ A549 cells exposed to 1 Gy show responses similar to 82-6 hTert cells following treatment with ATMi, ATRi and CHK2i (Appendix A). In addition, DNA-PKcsi, alone or in combination with ATMi and/or ATRi, shows similar responses as in 82-6 hTert cells (Appendix A). Likewise, A549 cells exposed to 4 or 10 Gy experience similar suppression by ATMi as 82-6 hTert cells (Appendix A), a similar involvement of ATR (Appendix A) and similar responses to combined ATMi + ATRi treatment (Appendix A). DNA-PKcsi enhances the G_1_ checkpoint in A549 cells exposed to 4 Gy (Appendix A), as well as the residual checkpoints measured in cells treated with ATMi (Appendix A), ATRi (Appendix A) or a combination of ATMi + ATRi (Appendix A). We conclude that the mechanism of regulation of the G_1_ checkpoint is similar in normal and cancer cell lines.

### 3.3. Activation of Cell Cycle and Checkpoint Proteins in Growth-Stimulated G_0_/G_1_ Cells

We analyzed the expression of cell cycle and checkpoint-related proteins in irradiated (4 Gy) and non-irradiated SD-derived G_0_ 82-6 hTert cells, either while maintained in G_0_, or after transfer to fresh growth medium. Cells maintained in G_0_ throughout the experiment show after irradiation robust stabilization of p53 that is maintained during the 24 h of observation (Figure 6a). This is interesting because non-cycling cells do not need p53 to suppress progression into the S phase and suggests that p53 exerts, under these conditions, functions beyond the suppression of cell cycle progression. When cells are released into the cell cycle but are not irradiated, p53 remains low during the 24 h of observation. After IR, however, stabilization of p53 is observed that is maintained up to 6 h, subsiding later (Figure 6a). This result suggests that the G_1_ checkpoint activated in G_0_ cells is p53 dependent.

Results in line with the expected cell cycle progression are obtained by analysis of cyclin D1. Cyclin D1 levels are low in G_0_ cells but show a small increase 12 and 24 h after exposure to IR (Figure 6a). After medium change, cyclin D1 levels increase as expected at 6 h and remain high up to 24 h. Exposure to 4 Gy attenuates this effect; this is likely a consequence of cell arrest in G_1_ by the checkpoint, and is in line with its degradation as a component of the G_1_ checkpoint [43]. CHK2 levels remain unchanged in non-irradiated or irradiated cells maintained in G_0_. In cells released into the cell cycle, CHK2 levels remain unchanged in the absence of IR, but are slightly reduced after release and exposure to 4 Gy (Figure 6a). Here again, phosphorylation of CHK2 after exposure to IR is observed at pT68 and pS516 (see above). The levels of CHK1 are low in unirradiated G_0_ cells and increase slowly but detectably 6–24 h after release and/or exposure to IR. The autophosphorylated form, CHK1-pS296, shows slight reduction after IR before release. It is strongly activated after release and remains equally active after IR.

On the other hand, the PI3K phosphorylated CHK1-pS345 form is nearly absent after IR before release but is strongly increased after release alone. Notably, in irradiated and released cells, CHK1-pS345 remains very low, likely a consequence of the checkpoint-mediated arrest of cells in G_1_/G_0_ (Figure 6a). This is evidence for the observed ATR activation. Overall, the results in Figure 6a document the activation of a p53-dependent checkpoint in G_0_ 82-6 hTert cells, demonstrate the expected activation of the cell cycle engine after transfer to fresh growth medium, and underpin the observed activation of ATR at high IR doses by the phosphorylation of CHK1.

### 3.4. Resection in G_1_-Phase Activates ATR at High IR Doses

Our recent work demonstrates striking parallels between ATR function in G_2_ checkpoint and resection at DSBs [17,18]. Therefore, after documenting a role for ATR in the G_1_ checkpoint at high IR doses, we explored similar associations in cells irradiated in G_0_/G_1_. Since resection is undetectable in cells irradiated in G_0_ (results not shown), we focused on resection analysis in G_1_ cells of actively growing cultures. To measure end resection, we quantitated RPA70 signal [44] in G_1_ cells using QIBC [17,18]. Figure 6b shows RPA70 signal intensity normalized to the levels of unirradiated 82-6 hTert cells. Exposure to 10 Gy causes clearly detectable resection that is markedly reduced by ATMi or ATRi, with combined treatment producing maximum effect (Figure 6b). At high IR doses, resection in G_1_ can also be analyzed by flow cytometry [17,18]. Appendix A shows the gates applied for this purpose, while Figure 6c summarizes the results obtained. Exposure to 20 Gy generates detectable resection 6 h later that is reduced by ATMi or ATRi. Here again, combined treatment with ATMi + ATRi eliminates resection. Thus, ATM/ATR-regulated resection at DSBs underpins ATR involvement in the G_1_ checkpoint at high IR doses.

## 4. Discussion

### 4.1. Variable Functional Integration of ATM, ATR and DNA-PKcs in Checkpoint Regulation

The results presented in the previous section extend and complement our previous studies on the regulation of DNA damage checkpoints by ATM, ATR and DNA-PKcs in S- and G_2_ phases of the cell cycle, for normal as well as for cancer cell lines, and confirm the importance of the IR dose in the orchestration of the underlying regulatory mechanisms [17,18,45]. Most notably, they emphasize the importance of the cell cycle phase in which DNA damage is inflicted, in the utilization of ATM, ATR and DNA-PKcs, to initiate the DSB signaling cascades that coordinately regulate the cell cycle machinery and the available DSB processing options, and indicate that this mechanism is not always altered in cancer cells.

Specifically, our results suggest that after exposure of G_1_- or G_0_-phase cells to low IR doses, the activation of the G_1_ checkpoint is exclusively mediated by ATM (Figure 7a) both in normal 82-6 hTert cells, as well as in the tumor cell line, A549. This is different from the activation of the G_2_ checkpoint in cells exposed to low IR doses in the S phase that is exclusively mediated by ATR (Figure 7b), and the activation of the G_2_ checkpoint in cells exposed to low IR doses in the G_2_ phase that shows epistatic regulation by ATM and ATR (Figure 7c). Not only ATM and ATR, but also DNA-PKcs is differentially utilized in checkpoint responses in different phases of the cell cycle. Thus, in cells irradiated in the G_2_ phase, DNA-PKcs by itself regulates checkpoint recovery, rather than activation, and in cells irradiated in the S phase, it provides this function together with ATM, while it has no detectable contribution to the G_1_ checkpoint regulation (Figure 7).

Notably, with increasing IR dose and the associated increase in the load of DSBs in the genome, the mechanism of the G_1_-checkpoint activation changes and becomes dependent also on ATR. Consistently, at high IR doses, the combined inhibition of ATM and ATR is required to completely suppress the checkpoint (Figure 7a). The cooperation between ATM and ATR in the regulation of the G_1_ checkpoint is similar to that observed in cells exposed to high IR doses in the G_2_ phase, where combined inhibition is required for complete suppression of checkpoint activation (Figure 7c). However, a difference between responses in the G_1_- and G_2_ phases is that while ATR is essential for low-dose checkpoint activation in the G_2_ phase, it plays no role in the low-dose activation of the checkpoint in the G_1_-phase. In addition, similar to cells irradiated in the S- or G_2_ phase, activation of ATR at high doses in the G_1_ phase is associated with resection at DSBs. This is similar to results in cells irradiated in the G_2_ phase, where resection is dependent on the activities of both ATM and ATR (Figure 7).

Collectively, the above results and those published earlier [17,18] indicate profound adaptability in the selective recruitment of ATM, ATR and DNA-PKcs to regulate checkpoint activation both in normal as well as in tumor cells (see Figure 7 for an overview). However, what underlying requirements of DNA damage processing are satisfied by this selectivity and adaptability? Checkpoint activation reflects the connection of DSB signaling with the cell cycle machinery to induce delays in the normal progression of cells through specific points in the cell cycle. It is widely accepted that the purpose of these delays is to optimize the processing of DNA damage—particularly the processing of DSBs. In the following section, we briefly discuss ATM-, ATR-, and DNA-PKcs-mediated connections between DSB processing and checkpoint activation in the light of known dependencies of DSB processing pathways throughout the cell cycle.

### 4.2. Contributions of ATM, ATR and DNA-PKcs to the Checkpoint and DSB Processing throughout the Cell Cycle in Normal and Cancer Cell Lines

When cells are irradiated in the G_2_ phase (Figure 7c), all DSB repair pathways can operate throughout the entire genome. At low IR doses, HR dominates [27] and links to the activation of resection that is epistatically regulated by ATM and ATR [18]. As a consequence, resection that is essential for the DSB repair pathway choice is directly coupled to the activation of the G_2_-phase checkpoint. Notably, analysis of the specific subset of DSBs that breaks chromosomes not only reveals the dominance of HR at low IR doses, but also suggests a mechanism suppressing c-NHEJ at low IR doses, as well as a switching mechanism to c-NHEJ as the IR dose rises [46,47]. Under these conditions, DNA-PKcs regulates resection [48], and suppression of its activity causes hyper-resection and checkpoint hyperactivation [18,49]. This is in line with reports, and actually explains why, sometimes, c-NHEJ appears not to be linked to checkpoints [18,50]. Thus, at low IR doses, ATM and ATR regulate resection and HR in the G_2_ phase and feed directly, through resection, into the activation of the G_2_-phase checkpoint. Not surprisingly, therefore, HR-deficient cells are deficient in the activation of the G_2_ checkpoint [51].

The role of DNA-PKcs in this choreography and the regulation of resection is incompletely characterized, but interesting hints are being reported [48]. DNA-PKcs is phosphorylated at multiple sites upon DNA damage by ATM [52,53], suggesting crosstalk and possibilities for functional integration. Loss of DNA-PKcs function hyperactivates ATM and p53 responses to DNA damage [54] and reduces ATM expression levels [55]. Other reports find, however, an attenuated G_2_-checkpoint response upon loss of DNA-PK function [56]. Evidently, more work is required to further elucidate the function of DNA-PKcs in cellular functions other than c-NHEJ.

At high IR doses that suppress HR, long before resection becomes also suppressed [27], ATM and ATR cooperate while also acting independently to regulate the G_2_ checkpoint [18]. The G_2_-checkpoint now assists resection-dependent pathways other than HR, as well as c-NHEJ, which, at high IR doses, becomes dominant [57]. The elucidation of details in the connections between checkpoint and DSB processing at high and low IR doses will require further investigations.

Irradiation of normal 82-6 hTert and tumor A549 cells in G_1_-phase presents a “simpler” situation (Figure 7a), as HR is fully compromised, owing to the absence of the sister chromatid. Other resection-dependent pathways are also compromised, because resection is overall compromised in the G_1_ phase [8,12]. Such circumstances allow c-NHEJ to dominate at low IR doses and to benefit from a checkpoint that is entirely dependent on ATM (Figure 1 and Figure 3). However, with increasing IR dose, the detectable resection (Figure 6) implies the activation of resection-dependent pathways, such as alt-EJ and SSA or even resection-dependent c-NHEJ [58], all of which may benefit from the activation of ATR and the cooperation between ATM and ATR in the activation of the G_1_ checkpoint (Figure 1 and Figure 4). Although resection remains undetectable at early times after growth stimulation and IR-exposure in G_0_ cells, we believe that as cells exit G_0_ and enter the G_1_ phase, they activate resection at levels equivalent to those of G_1_-phase cells. Here, again, details require further investigations.

Cells irradiated in the S phase also mount a checkpoint in the same phase that down-regulates replicon initiation [16]. This checkpoint generates delays of the order of minutes in the progression of the cell through the S phase, as opposed to the hours-long delays imposed by the checkpoints in the G_1_- and G_2_ phases. An S-phase cell has, at the same time, unreplicated genomic regions that resemble G_1_-phase chromatin and replicated genomic regions that resemble G_2_-phase chromatin. It represents, therefore, a hybrid situation, where both G_1_-like and G_2_-like responses are expected. How the coordinated action of ATM, ATR and DNA-PKcs regulates this aspect of checkpoint control is presently under investigation. Notably, however when S-phase-irradiated cells are allowed to complete DNA replication and enter the G_2_ phase, they activate a G_2_ checkpoint that is only ATR dependent (Figure 7b) [17]. The significance of this observation and the importance of cell cycle phase transition in this type of response requires further experiments.

Collectively, our results suggest the parallel recruitment of ATM, ATR and DNA-PKcs to DSB sites for the coordination of DSB processing in a cell-cycle-specific manner, which is directly linked to checkpoints. They emphasize that this regulation adapts to the DSB load and suggest uncharacterized mechanisms allowing cells to detect increases in the DSB load and translate them into changes in the DSB repair pathway choice and kinase crosstalk. We predict that further elucidation of the underpinning mechanisms will consolidate the integration of ATM, ATR and DNA-PKcs into a molecular mega machine coordinating DSB detection and DSB end-processing, and initiating DDR, including checkpoints, in a manner ensuring maximum genomic integrity in the face of necessities imposed by cell state and DSB form [2,17,18,27,59].

### 4.3. Putting Our Observations in the Context of the G_1_-Checkpoint Background

During the G_1_ phase, cells make critical decisions about their fate, including the optional commitment to replicate DNA and complete the cell cycle. If mitogens are available and the cellular environment is favorable for proliferation, a decision to enter the S phase is made at the so-called ‘restriction point’ in mid to late G_1_ [60]. The G_1_-phase checkpoint induced by DNA damage is mainly ascribed to p53 tumor suppressor protein [61,62]. Upon irradiation, cellular p53 becomes post-translationally modified, stabilized, and competent to induce the expression of genes required to halt cell-cycle progression or trigger programmed cell death [63] through a complex regulatory network [64,65].

Key upstream regulators of p53 activation are ATM, ATR and DNA-PKcs, as well as CHK1 and CHK2. Phosphorylation on Serine 20 of p53 by CHK2/CHK1 helps stabilize p53 by uncoupling it from the MDM2 ubiquitin ligase [66,67], while ATM/ATR-mediated phosphorylations of MDM2 at Ser 395 [68] and p53 at Ser 15 and some other residues interfere with nuclear export and help activate p53 [69,70]. DNA-PKcs interacts and phosphorylates p53, suggesting a role in the G_1_ checkpoint [71,72], and loss of function attenuates p53 downstream genes (p21 and MDM2) in response to DNA damage [73]. In addition to this p53-mediated G_1_ checkpoint that may last several hours, a more rapidly operating G_1_ checkpoint was reported, also initiated by ATM/ATR but rapidly targeting the CDC25A phosphatase for degradation to lock CDK2 in its inactive state and thus interfere with cell progression to the S phase [20,74]. A possible contribution of CDC25A degradation in the responses reported here will require further experiments.

Our results demonstrate that this activation of p53 is only dependent on ATM at low IR doses and that contributions by ATR become evident only at higher IR doses both in normal 82-6 hTert cell as well as in tumor A549 cells. The increased role of ATR in the G_1_ checkpoint with increasing IR dose relies on ATRIP-RPA binding on single-stranded DNA generated by end resection in the G_1_ phase. Other studies also suggest ATR activation in G_1_ phase cells and show that ATR inhibition selectively sensitizes G_1_-checkpoint-deficient cells to lethal premature chromatin condensation [75], and that CHK1 phosphorylation occurs in G_1_-synchronized U2OS cells [76].

Surprisingly, we failed to implicate CHK2 in the G_1_ checkpoint, which suggests that ATM-mediated phosphorylations of MDM2 are sufficient to maintain the activation of p53. Indeed, in our previous studies [17,18], we also failed to demonstrate a requirement for CHK2 in the G_2_-phase checkpoint despite its canonical activation at the protein level both in normal and in tumor cell lines. Furthermore, our results fail to convincingly implicate DNA-PKcs in the regulation of the G_1_ checkpoint. Future work will address these important questions.

In summary: The results of this report broaden our understanding of organization throughout the cell cycle and adaptation, with increasing IR dose, of the ATM/ATR/DNA-PKcs module to regulate checkpoint responses. They emphasize notable similarities and distinct differences between G_1_-, G_2_- and S-phase checkpoint regulation that may guide DSB processing decisions.

## Figures and Tables

**Figure 1 cells-11-00063-f001:**
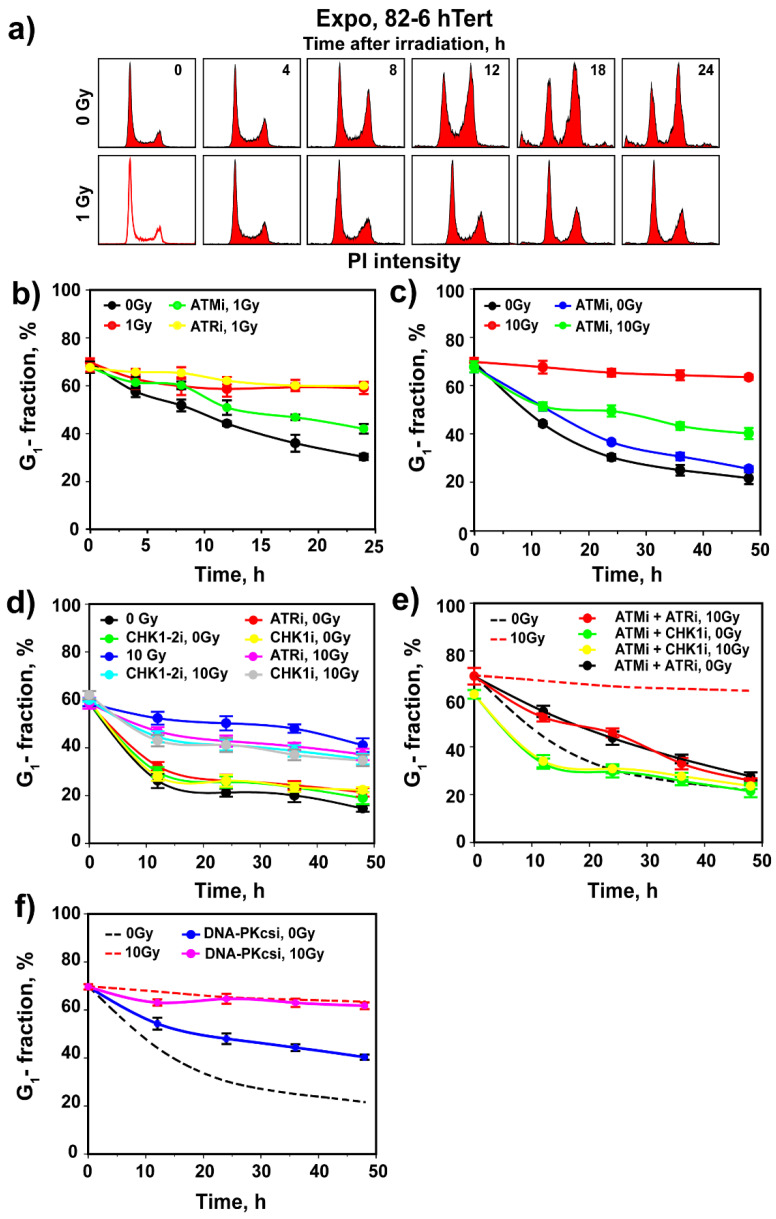
Assessment of G_1_ checkpoint in irradiated G_1_ cells of exponentially growing 82-6 hTert cell cultures. (**a**) Histograms showing cell cycle distribution as a function of time after exposure to 0 or 1 Gy of X-rays. Owing to the addition of nocodazole in the growth medium just before IR exposure, cell cycle progression causes a progressive enrichment in cells with G_2_/M DNA content. Note that this enrichment is delayed after IR, owing to the activation of several checkpoints in the cell cycle. The reduction in the fraction of cells with G_1_ content as a function of time reflects the progression of G_1_ cells into S phase. Note that IR, as a consequence of the activation of G_1_ checkpoint, delays the rate of reduction in G_1_ fraction. Experiments are reproduced 3 times and a representative one is shown. (**b**) Plot of G_1_ fraction as a function of time obtained from flow cytometry histograms such as those shown in a. The reduction measured in non-irradiated cells (black circles) reflects the rate of progression of G_1_ cells into S phase; the strong reduction observed in this rate in cells exposed to 1 Gy (red circles) reflects the activation of G_1_ checkpoint. This checkpoint is reduced after treatment with ATMi (green circles) but remains unchanged after treatment with ATRi (yellow circles). (**c**) As in (**b**) for cells exposed to 0 or 10 Gy and treated with ATMi as indicated. (**d**) As in (**b**) for cells exposed to 0 or 10 Gy and treated with ATRi, CHK1i, and CHK1-2i as indicated. (**e**) As in (**b**) for cells exposed to 0 or 10 Gy and treated with combinations of ATMi, ATRi and CHK1i as indicated. (**f**) As in (**b**) for cells exposed to 0 or 10 Gy and treated with DNA-PKcsi as indicated. Broken lines show results from c without data points to avoid congestion. Plotted in b–f is the mean and standard error (SE) from three independent experiments. See also Appendix A for more information.

**Figure 2 cells-11-00063-f002:**
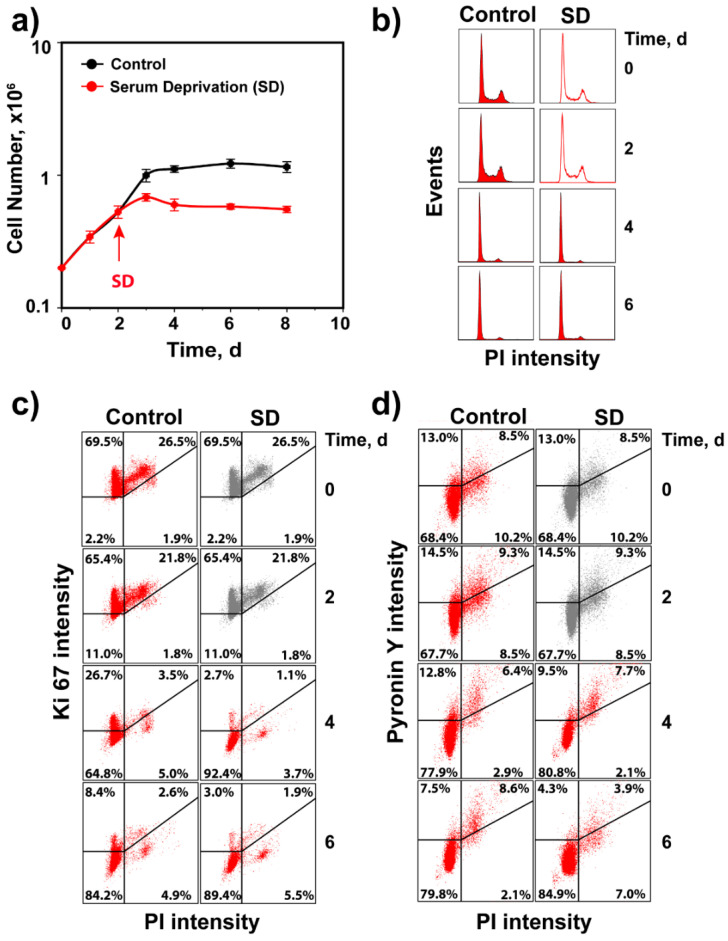
Validation of 82-6 hTert cell cultures for analysis IR-induced G_1_ checkpoint in cells irradiated in G_0_. (**a**) Proliferation of 82-6 hTert cells under normal growth conditions, as well as after transfer two days later to serum-free medium (serum deprivation, SD). Plotted is the mean and SE from three independent experiments. (**b**) Cell cycle distribution at the indicated times for cells growing as in A. Note the progressive enrichment in cells with G_1_/G_0_ content. (**c**) Characterization of cell cultures in A for Ki67 signal. (**d**) As in C for pyronin Y signal. In (**b**–**d**), experiments are reproduced 3 times, and one representative is shown.

**Figure 3 cells-11-00063-f003:**
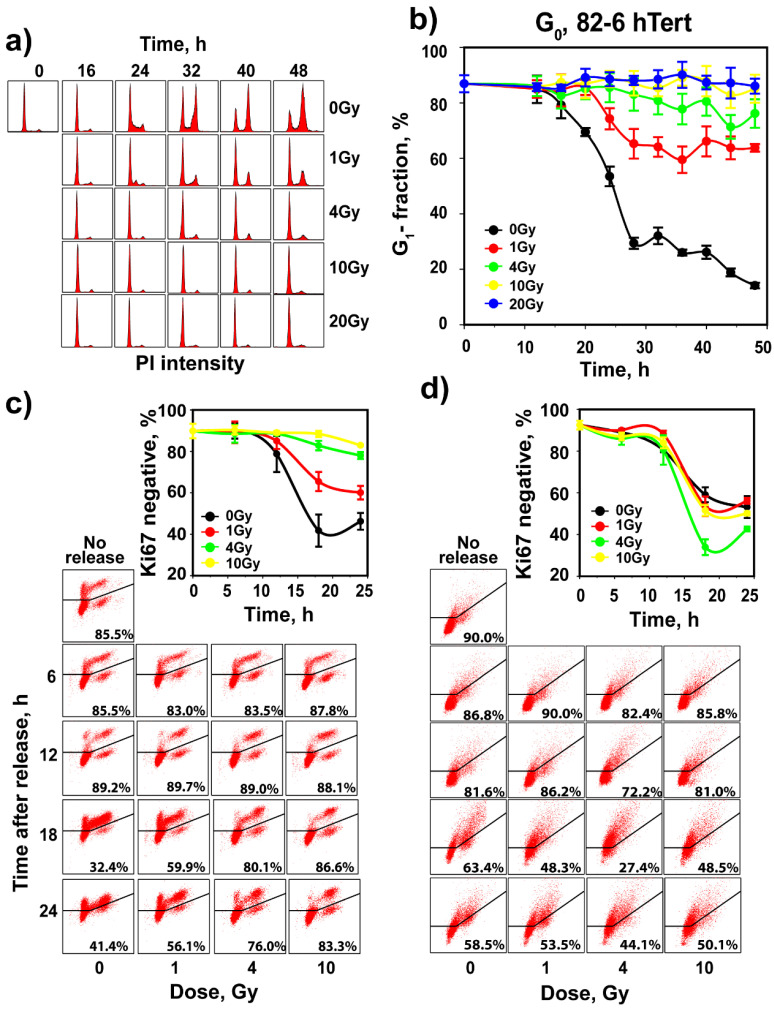
Assessment of G_1_ checkpoint in irradiated G_0_ cells of SD 82-6 hTert cell cultures. (**a**) Histograms showing cell cycle distribution as a function of time after exposure to different doses of X-rays. Cells are transferred to complete growth medium supplemented with nocodazole just before irradiation. Owing to the addition of nocodazole in the growth medium, cell cycle progression causes a progressive enrichment in cells with G_2_/M DNA content. The reduction in the fraction of cells with G_0_/G_1_ content as a function of time reflects the progression of G_0_ cells into the S phase. Here again, the reduction in this rate reflects the activation of G_1_ checkpoint. (**b**) As in Figure 1b for SD, 82-6 hTert cells exposed to the indicated doses of IR. (**c**) Analysis of Ki67 in the cell cultures analyzed in (**b**). (**d**) Analysis of pyronin Y in the cell cultures analyzed in (**b**). Plotted is the mean and SE from three independent experiments. See also Appendix A for more information.

**Figure 4 cells-11-00063-f004:**
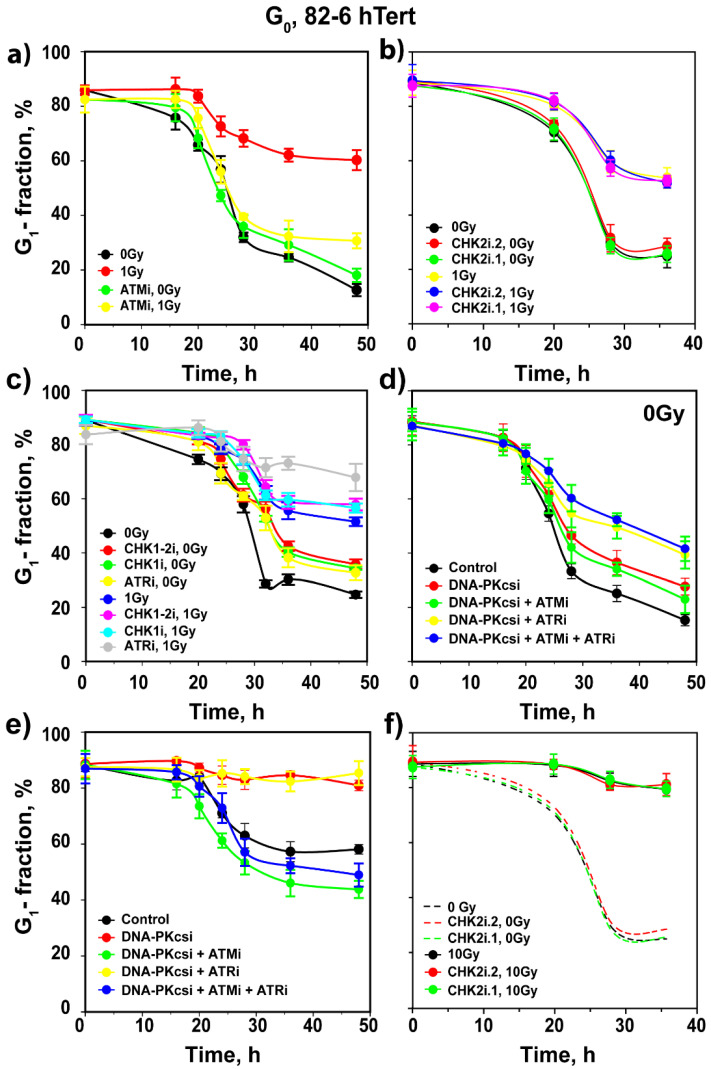
Assessment of the effects of ATM, ATR and DNA-PKcs inhibitors on the activation of G_1_ checkpoint in irradiated G_0_ cells of SD 82-6 hTert cell cultures. (**a**) As in Figure 1b for cells exposed to 0 or 1 Gy in the presence of ATMi as indicated. (**b**) As in (**a**) for cells exposed to CHK2 inhibitors as indicated. (**c**) As in (**a**) for cells exposed to CHK1 inhibitors as indicated. (**d**) As in (**a**) for unirradiated cells treated with DNA-PKcsi in the presence or absence of ATMi and or ATRi. (**e**) As in (**d**) for cells exposed to 1 Gy of X-rays. (**f**) As in panel (**a**), for cells exposed to 0 or 10 Gy in the presence or absence of a CHK2 inhibitor as indicated. Broken lines show results from (**b**) without data point to avoid congestion. Plotted is the mean and SE from three independent experiments. See also Appendix A for more information.

**Figure 5 cells-11-00063-f005:**
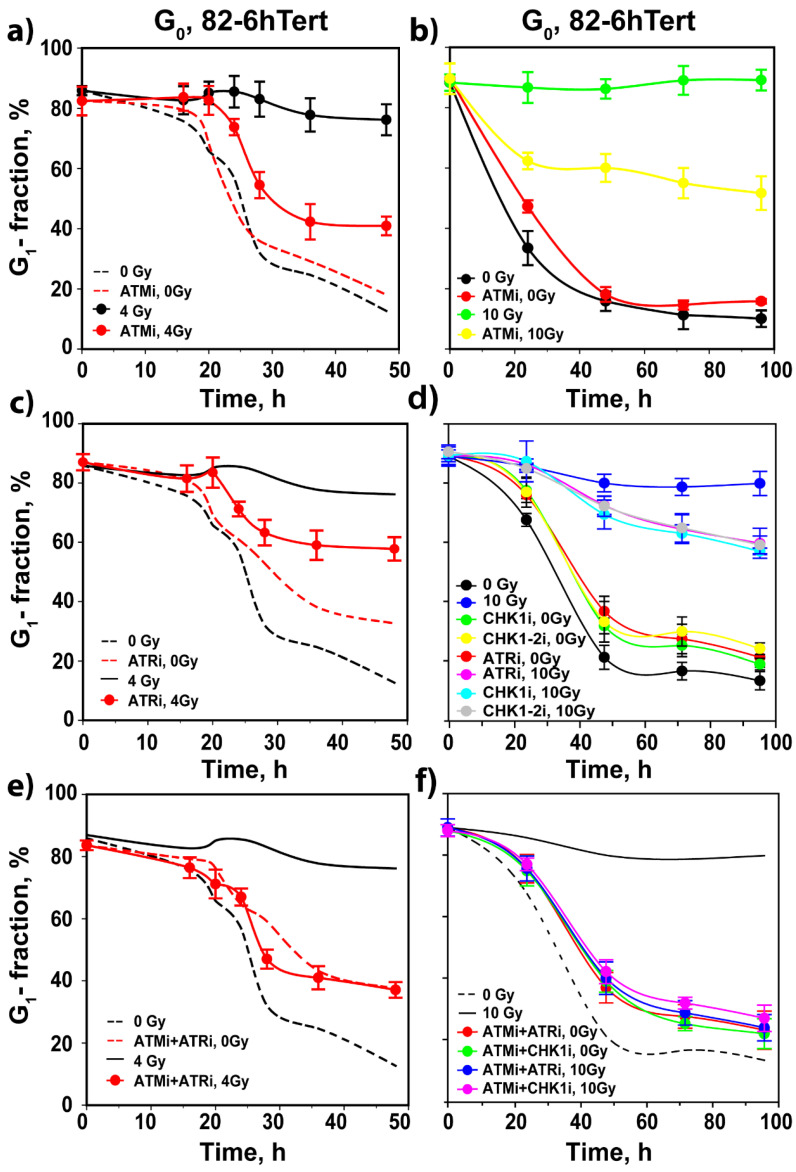
Assessment of the effects of combined treatment with inhibitors of ATM, ATR and DNA-PKcs on the activation of the G_1_ checkpoint in irradiated G_0_ cells of SD 82-6 hTert cell cultures. (**a**) As in Figure 1b for cells exposed to 0 or 4 Gy in the presence or absence of ATMi as indicated. (**b**) As in (**a**) for cells exposed to 10 Gy. (**c**) As in (**a**) for cells exposed to ATRi. (**d**) As in (**b**) for cells exposed to ATRi, CHK1i, and CHK1-2i. (**e**) As in (**a**) for cells exposed to combined ATMi and ATRi. (**f**) As in (**b**) for cells exposed to the indicated combinations of ATMi, ATRi and CHK1i. Broken and solid lines without data points show results from Figure 4, Figure 5a or Figure 5d; symbols are omitted to avoid congestion. Plotted is the mean and SE from three independent experiments. See also Appendix A for more information.

**Figure 6 cells-11-00063-f006:**
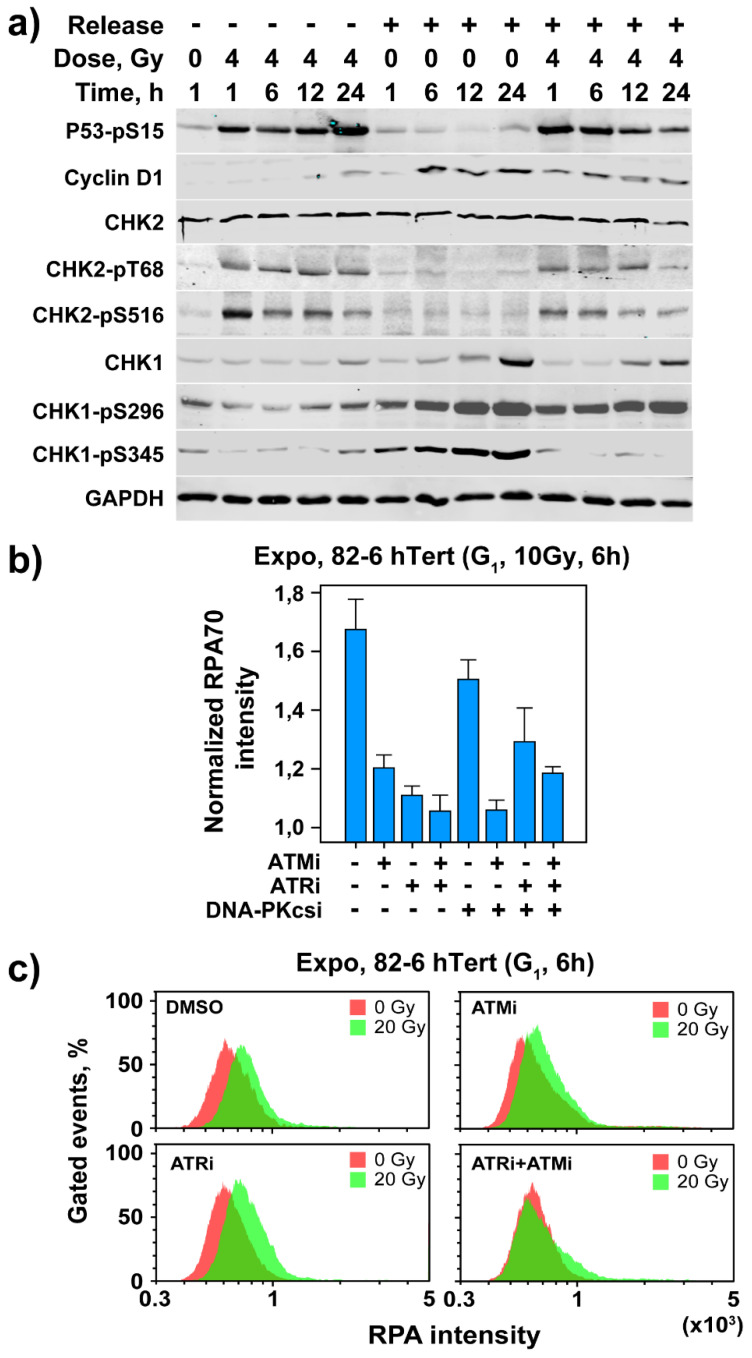
Assessment of cell cycle and checkpoint proteins in irradiated and non-irradiated G_0_ cells of SD 82-6 hTert cell cultures with or without transfer to fresh growth medium; analysis of resection at DSBs in G_1_ cells of exponentially growing cultures. (**a**) Levels of indicated proteins as a function of time after exposure to 0 or 4 Gy (see text for details). Experiments are reproduced 3 times and a representative one is shown. (**b**) Analysis of DNA end resection at DSBs in G_1_ phase after exposure to 10 Gy in the presence or absence of ATMi, ATRi and DNA-PKcsi at the indicated combinations using QIBC as described under “Material and Methods” and RPA70 intensity as endpoint. Plotted is the mean and SE from three independent experiments. See also Appendix A for more information. (**c**) Analysis of DNA end resection at DSBs in G_1_ phase after exposure of cells to 20 Gy in the presence or absence of ATMi and/or ATRi as indicated, using flow cytometry as described under “Material and Methods” and RPA70 signal intensity as endpoint. Experiments are reproduced 3 times and a representative one is shown.

**Figure 7 cells-11-00063-f007:**
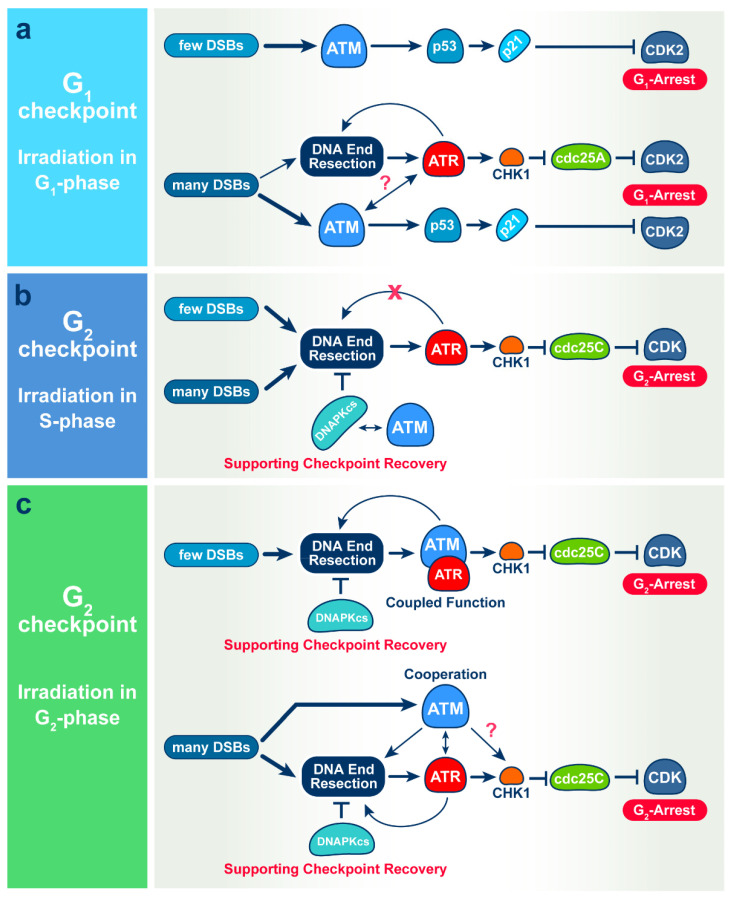
Mechanisms of IR-induced checkpoints in G_1_ (**a**), S (**b**) and G_2_ (**c**) phase of the cell cycles. See text for details.

## Data Availability

Not applicable.

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
