# Peer review of "Shift in G1-Checkpoint from ATM-Alone to a Cooperative ATM Plus ATR Regulation with Increasing Dose of Radiation"

_cells, 2021, doi:10.3390/cells11010063_

Round 1
Reviewer 1 Report
In the present manuscript “Shift in G1-checkpoint from ATM-alone to a cooperative ATM plus ATR regulation with increasing dose of radiation”, the authors explore the organization throughout the cell cycle and the adaptation with increasing IR-dose, of the ATM/ATR/DNA-PKcs module to regulate checkpoint responses.
- In general, the submitted manuscript is very well structured and with very carefully written, without major spelling or grammatical errors.
- I think that a summary in the form of a graphical abstract will help the readers to easily understand the main ideas that authors wanted to cover.
- Introduction provides important and updated theoretical considerations of the different topics necessary for the all understanding of the paper.
- Some comments that should be explained or addressed somehow:
- Lines 112: CtIP abbreviation is not previously defined.
- Line 36: Consider to change “for more and good”
- Line 132: energy of X-ray must be in KeV instead of kV
- Line 131: In the irradiation process, was the uniformity of dose distribution guaranteed? How? Please describe.
- Details on the irradiation procedure are missing in Methods section. Which exposed doses were used? Authors just refer to dose rates, but do not refer to absorbed doses tested for each experiment.
- References of antibodies or suppliers should be provided. At least in “Pyronin and Ki67 Staining” they are not specified.
- Methodologies are written both in the past and in the present. I suggest to revise and harmonize in the past.
- Statistical analysis is missing in all data.
- Line 170: If you use 10uM of final concentration of the ATM inhibitor, you will also have DNA-PKcs inhibition (Ic50=2,5uM). Isn’t it right? The same will happen for the ATRi. How do guarantee a specific inhibition?
- Please define the abbreviation the first time they appear in the text. Example: PBS is defined in line 200, but already appeared before. SD abbreviation is double-used, as serum deprivation and as standard deviation. Please correct.
- Line 219: I suggest to change the section title, considering western blot already includes electrophoresis.
- Lines 256-257: please rephrase.
- Line 264: “into”
- Why are so many results shown as histograms (just with one representative image) instead of more complete graphs that could reflect are the results obtained in the independent experiments?
- Figure 3: Correct “negetive”.
- Why dashed lines in different graphs (e.g. 1E, 1F, 4F) do not present means and standard deviations?
- Line 579: “such as”
- Beyond graphical abstract, figures illustrating the major conclusions pointed out and so well discussed by the authors will really improve the quality and the dissemination of the paper. g.: it would be important to schematically represent the similarities and distinct differences between G1-, G2- and S-phase checkpoint regulation that may guide DSB processing decisions.
- The authors use two types of cell lines: one non-tumoral (82-6 hTert) and one cancer cell line (A549), however, they never discuss why to use both. In fact, it is not clear why A549 cells were included in the study, since very few results were included and the differences in terms of checkpoints regulation with increasing IR doses in non-tumor cells compared to cancer cells is not an issue addressed in the discussion. Knowing in advance that cell cycle regulation in really implicated in the hallmarks of cancer, the response to IR will not be the same in normal cells and cancer cells. For these reasons, I suggest to include A549 results in a separate manuscript.
Overall, this manuscript is worth of publication in Cells journal with major revisions and clarifications.
Reviewer 2 Report
Li et al investigated the potential role of ATM, ATR and DNA-PKcs phosphoinositide 3-kinase related kinases in the development of G1 block after low and high dose irradiations. Cells were arrested in the G0/G1 phase of the cell cycle by serum deprivation and then irradiated by 1 or 10 Gy x-rays. Irradiating the cells in G0/G1 phases induces a G1 phase cell cycle block. The role of the ATM, ATR and DNA-PKcs in the development of the G1 phase block was investigated by adding agents specifically inhibiting the effects of these kinases.
The authors report here that after irradiation of the cells with 1 Gy ATM is the sole regulator of the G1 block. At 10 Gy ATR is also contributes to the dominating role of ATM. The authors conclude that after irradiation of the cells in G0/G1 phase DNA-PKcs exhibits no role in the development of G1 checkpoint.
The paper is well-written, the experiments are well-designed and the data clearly support the conclusions. The results contribute strongly to our better understanding of the regulation of radiation-induced double-strand break DNA repair.
Minor concerns
- The reviewer did not find the link to the supplementary figures.
- Why do you call 1 Gy a low dose?
Author Response
Reviewer 2
General Comments
- Li et al investigated the potential role of ATM, ATR and DNA-PKcs phosphoinositide 3-kinase related kinases in the development of G1 block after low and high dose irradiations. Cells were arrested in the G0/G1 phase of the cell cycle by serum deprivation and then irradiated by 1 or 10 Gy x-rays. Irradiating the cells in G0/G1 phases induces a G1 phase cell cycle block. The role of the ATM, ATR and DNA-PKcs in the development of the G1 phase block was investigated by adding agents specifically inhibiting the effects of these kinases.
The authors report here that after irradiation of the cells with 1 Gy ATM is the sole regulator of the G1 block. At 10 Gy ATR is also contributes to the dominating role of ATM. The authors conclude that after irradiation of the cells in G0/G1 phase DNA-PKcs exhibits no role in the development of G1 checkpoint.
The paper is well-written, the experiments are well-designed and the data clearly support the conclusions. The results contribute strongly to our better understanding of the regulation of radiation-induced double-strand break DNA repair.
We thank the Reviewer for the positive evaluation of our work.
Minor concerns
- The reviewer did not find the link to the supplementary figures.
We sincerely apologize for the complication. Because we had uploaded all Supplementary Material related to the paper at the Journal site, we assume that the problem was specific at the Reviewer’s end.
- Why do you call 1 Gy a low dose?
We thank the Reviewer for pointing out this important issue. Indeed, the characterization of a radiation dose as low or high depends on actual purpose or use, and is in some ways arbitrary. For example, in radiation protection, 1 Gy is a high dose! In our case the definition is based on biological responses to DNA damage and shifts in DSB repair pathway balance that we have detected. In the revision we state explicitly that the definition is arbitrary, explain the rationale we used and define doses below 2 Gy as low and doses above 2 Gy as high.
Round 2
Reviewer 1 Report
The authors addressed the majority of suggestions made.
The only issue I think is very important and that was not sufficiently addressed respects to point 19 of the Author response to report 1:
19. We appreciate and understand the background logic of the Reviewer’s suggestion. Yet, in our past work, when we described aspects of checkpoint regulation, we went long ways in showing that our observations are valid in more than one cell lines. In the present paper we continue on this line of experimentation. At the same time, the editors of Cells indicate in their Instructions to Authors: “the use of more than one cell line is encouraged, and when this is not possible, authors need to justify their use of a single cell line”. For all these reasons, we would like to opt for inclusion in the present paper of the results with A549 cells. However, the Reviewer correctly points out that if included, these results need to be discussed and compared more extensively. Indeed, in the revised manuscript, we expand on these points.
To present few results with an alternative cell line just to comply with the Cells journal instructions is not sufficient, but in fcat they may be considered if well discussed. However, those results were poorly discussed and in the revised document I just saw a small addition (lines 463-465) regarding this "We conclude that the mechanism of regulation of the G1-checkpoint is similar in normal and cancer cell lines.", which I think is not enough. So, I suggest the authors to improve this issue.
Author Response
Response to Reviewer 1:
- To present few results with an alternative cell line just to comply with the Cells journal instructions is not sufficient, but in fact they may be considered if well discussed. However, those results were poorly discussed and in the revised document I just saw a small addition (lines 463-465) regarding this "We conclude that the mechanism of regulation of the G1-checkpoint is similar in normal and cancer cell lines.", which I think is not enough. So, I suggest the authors to improve this issue.
Response: Thanks a lot for the suggestion and for pointing out this omission. We added in the discussion several explicit mentions that the results obtained have been confirmed in the normal 82-6 hTert cells, as well as in the A549 tumor cell line. We hope that these changes satisfy the remaining concern.